# Multiple conserved cell adhesion protein interactions mediate neural wiring of a sensory circuit in *C. elegans*

**Byunghyuk Kim[1]\*, Scott W Emmons[1,2]\***

[1]Department of Genetics, Albert Einstein College of Medicine, Bronx, United States; [2]Dominick P Purpura Department of Neuroscience, Albert Einstein College of Medicine, Bronx, United States

**Abstract** Nervous system function relies on precise synaptic connections. A number of widely-conserved cell adhesion proteins are implicated in cell recognition between synaptic partners, but how these proteins act as a group to specify a complex neural network is poorly understood. Taking advantage of known connectivity in *C. elegans*, we identified and studied cell adhesion genes expressed in three interacting neurons in the mating circuits of the adult male. Two interacting pairs of cell surface proteins independently promote fasciculation between sensory neuron HOA and its postsynaptic target interneuron AVG: BAM-2/neurexin-related in HOA binds to CASY-1/calsyntenin in AVG; SAX-7/L1CAM in sensory neuron PHC binds to RIG-6/contactin in AVG. A third, basal pathway results in considerable HOA-AVG fasciculation and synapse formation in the absence of the other two. The features of this multiplexed mechanism help to explain how complex connectivity is encoded and robustly established during nervous system development.

DOI: https://doi.org/10.7554/eLife.29257.001

**\*For correspondence:**
byunghyuk.kim@einstein.yu.edu
(BK);
scott.emmons@einstein.yu.edu
(SWE)

**Competing interests:** The authors declare that no competing interests exist.

## Introduction

A cardinal feature of the nervous system is its massively large number of specifically connected cells. The effort to reconstruct synaptic connectivity in the nervous system of the nematode *Caenorhabditis elegans* was undertaken in part to study the genetic specification of nervous system structure (*Brenner, 1974*). One central component of this process must consist of a molecular mechanism during development of synapses for specific cell-cell recognition. Knowledge of the complete connectome places certain requirements and constraints on the possible nature of such a mechanism (*Durbin, 1987*; *Emmons, 2016*; *Jarrell et al., 2012*; *White, 1985*; *White et al., 1983*; *White et al., 1986*). As one example, even in this small (<400 neurons) nervous system, which comprises a sparse neural network, the number of different chemically and electrically coupled cell pairs is large, over 6000. It is therefore likely that a combinatorial mechanism of some kind is necessary to encode this complexity with a reasonable number of genes.

The mechanism has long been thought to depend on the molecular selectivity of ligand-receptor interactions between cell surface proteins (*Sperry, 1963*). In *C. elegans*, each neuron connects to multiple (10 or more) other cells — neurons, muscles, and other end organs. Since these sets of target cells intersect, each cell must express multiple such recognition molecules, a different set defining each cell type. The cellular architecture of the nervous system is reproducible, with each neuron process running in a small number of conserved neighborhoods, where it makes *en passant* chemical and gap junction synapses. As a neuron can only synapse onto a cell with which it is in contact, cell recognition events establishing and stabilizing neighborhoods likely play an important role in specifying the connectome (*White, 1985*; *White et al., 1983*).

Quantitative analysis showed the strengths of the connections made to different targets varies continuously over a wide range (*Jarrell et al., 2012*). This suggests a mechanism that allows for a genetically-specified probability of synapse formation for each pair of cell types. One way to achieve this would be to employ multiple, parallel, pathways acting additively. A multiply over-determined system allows both for robustness and evolvability. It may explain why the overall mechanism remains obscure in spite of decades of forward genetic screens focused on the nervous system and behavior. Multiple additive signals might also account for another feature of the *C. elegans* nervous system, the prevalence of polyadic chemical synapses (69% in hermaphrodites and 66% in males). These might be accounted for if coincident signals from several appropriate postsynaptic target cells increase the probability of presynapse formation. Prior studies in *C. elegans* have implicated interactions with local, albeit non-neuronal, cells in increasing the probability of synapse formation (*Colón-Ramos et al., 2007*; *Shen and Bargmann, 2003*).

The *C. elegans* genome contains 106 genes encoding transmembrane or secreted proteins with extracellular protein interaction domains — a structure expected for neural cell-surface recognition molecules (*Hobert, 2013*). Many of these presumptive cell adhesion genes are conserved in higher animals and their counterparts have been shown to be involved in synapse formation or stabilization (*de Wit and Ghosh, 2016*).

We initiated our study by determining the expression of this set of 106 genes in the neurons and muscles of the neural network for male mating behavior (in preparation). Based on this expression analysis, here we have analyzed the functions of the cell adhesion genes expressed by a male-specific sensory neuron (HOA) and two of its postsynaptic targets, a sex-shared interneuron (AVG) and a pair of sex-shared sensory neurons (PHCL and PHCR). The neurons make up a part of the circuits for copulatory behavior in the male tail and are necessary for normal male copulatory behavior (*Jarrell et al., 2012*; *Liu and Sternberg, 1995*). We find that the mechanism promoting connectivity of HOA to AVG has precisely the properties proposed above. The neurons express multiple cell adhesion genes. Interactions between a subset of these, two pairs of conserved cell adhesion genes, act together with a basal pathway in three independent pathways that function additively to promote fasciculation of the three neurons, leading to synapse formation.

## Results

### Two cell adhesion genes, *casy-1* and *rig-6*, are required for patterning of a synaptic vesicle protein and axon fasciculation

For in-depth analysis of circuit formation, we focused on a male-specific circuit that consists of a male-specific sensory neuron, HOA, a sex-shared sensory/interneuron, PHC (a left/right pair of neurons, PHCL and PHCR), and a sex-shared interneuron, AVG (*Figure 1A*). This circuit has two properties: (1) the axons of the three neurons are highly associated with each other to make a bundle within the male pre-anal ganglion (*Figure 1B and C*); (2) in this region, the HOA axon forms *en passant* dyadic chemical synapses onto AVG and PHC (*Figure 1B and D*). (In one animal reconstructed by electron microscopy, there are ~11 clusters of HOA presynaptic densities comprised of 44 synapses with AVG and 46 synapses with PHC (*Supplementary file 1*). The PHC axon also forms dyadic synapses onto AVG and HOA.) Outside of this region, HOA has many other synaptic targets including male-specific neurons, PVZ, PCA, and some ray sensory neurons (WormWiring: http://wormwiring.org).

AVG, which is mostly postsynaptic to the other neurons, expressed three neural cell adhesion genes in our expression dataset, *casy-1*, *rig-6*, and *sax-3* (*Figure 1E*). To test whether these genes function for circuit formation, we visualized HOA synaptic output using a fluorescently tagged protein of presynaptic vesicles, mCherry::RAB-3, as a proxy for synaptic connections between HOA and AVG (*Figure 1—figure supplement 1*) and examined the effects of mutations on the distribution of the presynaptic puncta. We used null mutant alleles, *casy-1(tm718)*, *rig-6(gk438569)* and *sax-3 (ky123)*. For *rig-6*, we also used a hypomorphic *rig-6(ok1589)* allele, which is predicted to affect three out of four isoforms of RIG-6. (Throughout the paper the hypomorph was used because of its ease of genetics. Its phenotypes were the same as the null [see *Figure 1—figure supplement 3* and *Figure 2—figure supplement 2*]). In wild type, multiple RAB-3 puncta were evenly spaced throughout the HOA process (*Figure 1F*). All mutations disrupted normal puncta distribution: mutants for *casy-1*

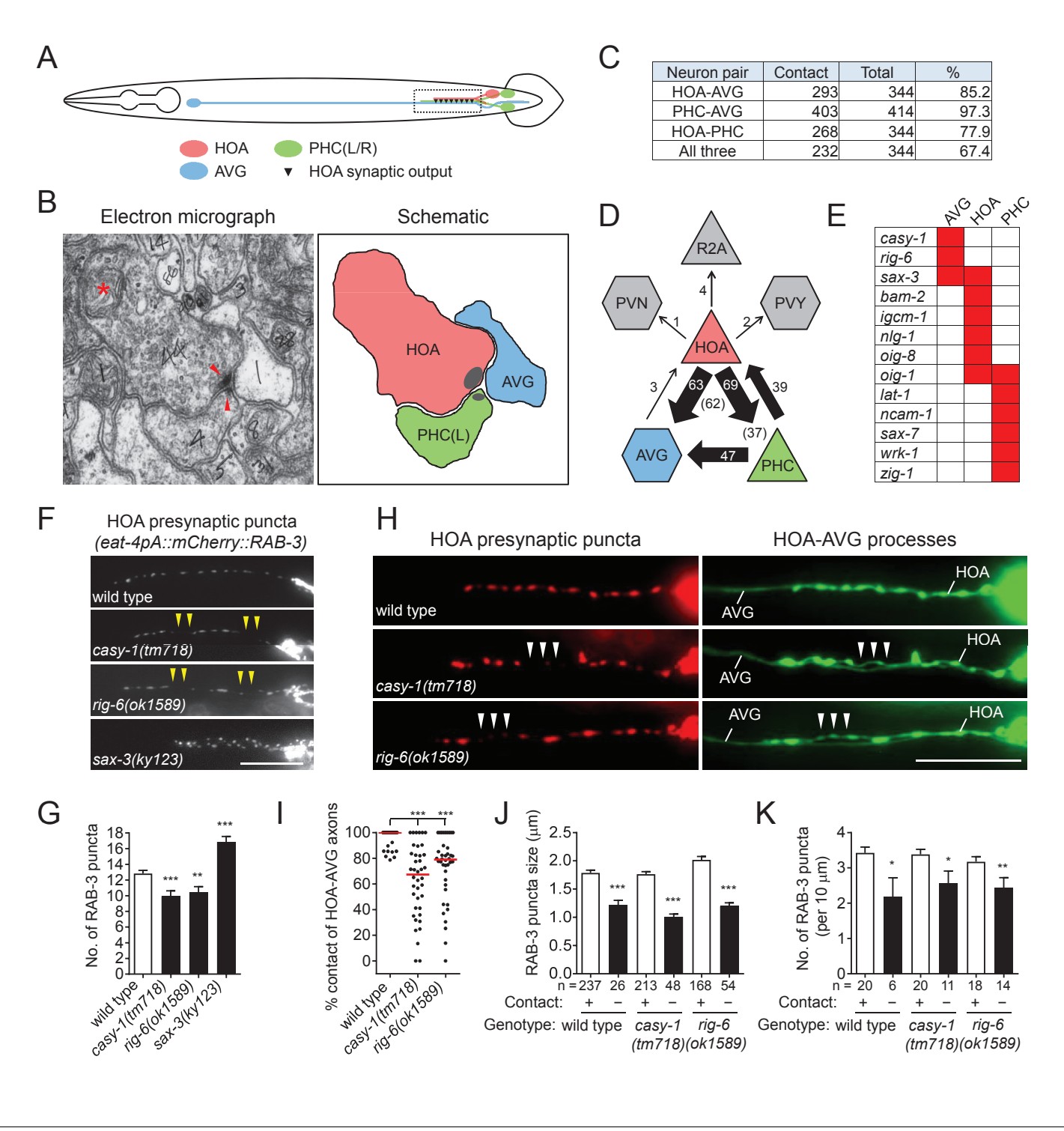

**Figure 1.** Cell adhesion protein genes *casy-1* and *rig-6* are required for axon fasciculation. (**A**) Schematic of the position of cell bodies and axons of three neurons HOA (red), AVG (blue) and a pair of PHC (green) in the male (ventral view). The dashed box indicates the axons and HOA synaptic output analyzed in (**C and D**) and imaged in (**F and H**). (**B**) Electron micrograph showing the axons and synapses of HOA, PHC and AVG. Arrowheads indicate presynaptic density of dyadic synapses of HOA and PHC. Asterisk indicates a mitochondrion. In the schematic, axons are colored as in (**A**), and presynaptic density is indicated as gray. (**C**) The number of electron micrograph sections for axon-axon contact between neuron pairs. N2Y series were analyzed from section #13906 to #14249, except for the PHC-AVG (from #13836 to #14249) (WormWiring: http://wormwiring.org). See also *Supplementary file 1*. (**D**) Synaptic output of HOA and the connectivity of HOA, AVG and PHC in the boxed region shown in (**A**). (HOA makes

*Figure 1 continued*

additional connections outside this region.) Synaptic weight determined by electron micrograph section numbers is indicated. The number of sections that contain dyadic synapses (HOA > AVG,PHC or PHC > HOA,AVG) is indicated in parenthesis. (E) Expression of transcriptional reporters for neural cell adhesion genes in the three neurons. One hundred out of 106 neural cell adhesion genes (94%) have been examined and the genes having expression in the three neurons are shown. (F) Distribution of a mCherry-tagged presynaptic marker RAB-3 in HOA axon of wild type or indicated mutants. Arrowheads indicate gaps between the presynaptic puncta. (G) Number of mCherry::RAB-3 puncta in mutants was counted and compared to wild type (n = 30). Error bars are SEM. (H) HOA presynaptic puncta (mCherry::RAB-3; red) were simultaneously visualized with GFP-labeled HOA and AVG axons (green) in wild type, or *casy-1* or *rig-6* mutant animals. Arrowheads indicate the gap region containing smaller or fewer presynaptic puncta. (I) Percentage of axon-axon contact between HOA and AVG in wild type or mutant animals (n = 40). Each dot represents individual animal. Red bar represents the median. (J and K) The mCherry::RAB-3 puncta size (J) or number (K) was measured and compared in the contacting and non-contacting axonal segments between HOA and AVG for the indicated genotypes. The number of the puncta (J) or of axon segments (K) analyzed is indicated below each column. Error bars are SEM. Scale bars, 20 μm. *p<0.05; **p<0.01; ***p<0.001 (by Mann-Whitney test). For the data and statistics, see *Figure 1—source data 1*.

DOI: https://doi.org/10.7554/eLife.29257.002

The following source data and figure supplements are available for figure 1:

**Source data 1.** Source data for *Figure 1*, *Figure 1—figure supplement 1*, and *Figure 1—figure supplement 2*.

DOI: https://doi.org/10.7554/eLife.29257.006

**Figure supplement 1.** Visualizing HOA presynapses.

DOI: https://doi.org/10.7554/eLife.29257.003

**Figure supplement 2.** Extra branching of HOA in *sax-3(ky123)* mutants.

DOI: https://doi.org/10.7554/eLife.29257.004

**Figure supplement 3.** Phenotypes of *rig-6(gk438569)* mutants.

DOI: https://doi.org/10.7554/eLife.29257.005

and *rig-6* often exhibited gaps containing fewer or smaller puncta, whereas in mutants lacking *sax-3*, the number of RAB-3 puncta was increased due to extra branching of the HOA axon (*Figure 1F and G*, and *Figure 1—figure supplements 2* and *3*). SAX-3/Robo is a transmembrane protein with immunoglobulin (Ig) domains and is known to function in axon guidance of multiple types of neurons (*Zallen et al., 1999*). Thus it is likely that the extra branching phenotype we observed is one of the axon guidance defects in *sax-3* mutants, which we do not pursue further here.

GPI-anchored RIG-6/contactin belongs to the Ig superfamily and its loss leads to axon guidance defects of sensory and motorneurons in *C. elegans* (*Katidou et al., 2013*). CASY-1/calsyntenin is a cadherin domain-containing transmembrane protein. One of its mammalian homologs, calsyntenin-3, has been shown to induce in vitro presynapse differentiation when overexpressed postsynaptically (*Pettem et al., 2013*). As cell contact may be necessary for synapse formation, it was possible that the gap phenotype we observed in *casy-1* and *rig-6* mutants originated from defasciculation of HOA from its target axons. To test this possibility, we examined axon fasciculation between HOA and AVG, along with the HOA presynaptic puncta distribution in the mutants. Indeed, in *casy-1* and *rig-6* mutants, the HOA and AVG axons were frequently detached from each other (*Figure 1H and I*). Moreover, the axonal segments of HOA detached from AVG had smaller and fewer presynaptic puncta than those contacting AVG (*Figure 1H,J and K*). This phenotypic association of axon fasciculation and presynaptic pattern was not due to mutational effects, because in wild type animals we also observed, even though rarely, that detached axon segments contained smaller and fewer puncta (*Figure 1J and K*). Therefore, the HOA presynapse puncta pattern is tightly associated with contacts between HOA and AVG axons, and the gap phenotype we observed in mCherry::RAB-3 labeled presynaptic structures originates most likely from frequent HOA-AVG defasciculation in mutant animals.

To further examine axon fasciculation among the three associated neurons, HOA, AVG and PHC, in *casy-1* and *rig-6* mutants, we generated transgenic animals whose neurons were labeled with mCherry, TagBFP and GFP, respectively (*Figure 2A*). Then we used fluorescence images to measure fasciculation or defasciculation between the axons. This measurement was sensitive enough to detect subtle defasciculation events: in the wild type animal reconstructed by electron microscopy, the PHCL and PHCR axons, for example, are separated over 30% of their lengths by only one-neurite distance, and similar measurement of defasciculation was observed using fluorescence images (*Figure 2—figure supplement 1*). We found that the axon fasciculation defects of *casy-1* and *rig-6* mutants were distinguishable from each other: in mutants lacking *casy-1*, the HOA axon was

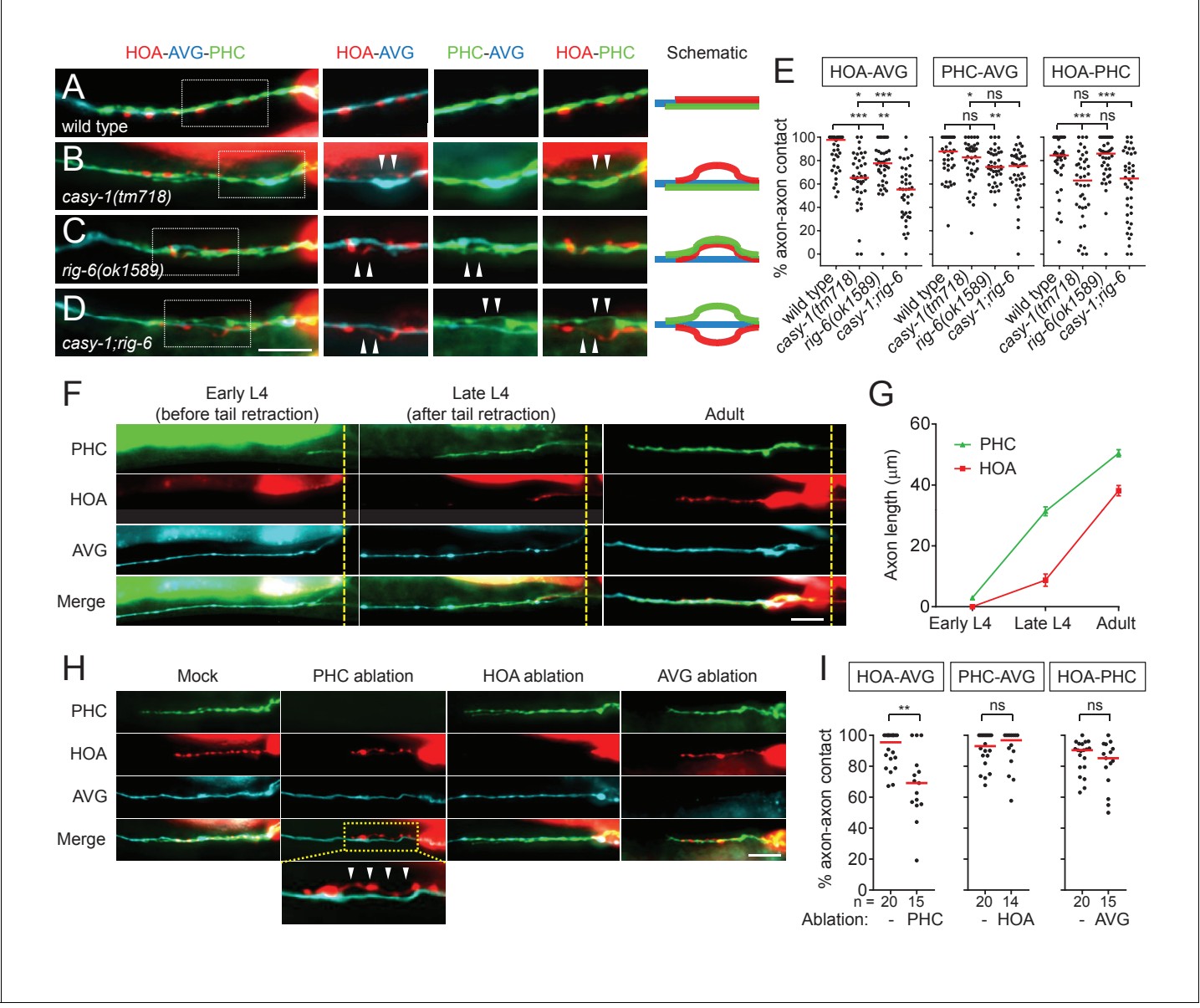

**Figure 2.** Axon fasciculation defects of *casy-1* and *rig-6* are distinguishable. (A–D) The axons of the three neurons were individually labeled with wCherry (HOA; red), TagBFP (AVG; blue), and GFP (PHC; green) in wild type (A), *casy-1(tm718)* (B), *rig-6(ok1589)* (C), or *casy-1;rig-6* double mutants (D). Axon fasciculation between each neuronal pair in the dashed box region and schematic of axon fasciculation are shown on the right. Arrowheads indicate the region where two axons are detached from each other. (E) Percentage of axon-axon contact between each neuronal pair in wild type or mutant animals (n = 40). Each dot represents individual animal. Red bar represents the median. (F) Developmental timing of axon extension of the three neurons in males. Each developmental stage was determined by the tail morphology of the male. Dashed lines indicate the posterior end of the intestine. (G) Axon length was measured from the point where the AVG axon makes a turn in the pre-anal ganglion (n = 15). Error bars are SEM. (H) Axon fasciculation of the three neurons in mock-, PHC-, HOA-, or AVG-ablated animal. Arrowheads indicate the region where HOA and AVG axons are detached from each other. (I) Percentage of axon-axon contact between each neuronal pair in mock-, PHC-, HOA-, or AVG-ablated animals. The number of animals analyzed is indicated. Scale bars, 20 μm. *p<0.05; **p<0.01; ***p<0.001; ns, not significant (by Mann-Whitney test). For the data and statistics, see *Figure 2—source data 1*.

DOI: https://doi.org/10.7554/eLife.29257.007

The following source data and figure supplements are available for figure 2:

**Source data 1.** Source data for *Figure 2* and *Figure 2—figure supplement 2*.
DOI: https://doi.org/10.7554/eLife.29257.010

**Figure supplement 1.** Comparison of PHCL-PHCR defasciculation in electron micrographs and fluorescence images.
DOI: https://doi.org/10.7554/eLife.29257.008

*Figure 2 continued on next page*

*Figure 2 continued*

**Figure supplement 2.** Phenotypes of *rig-6(gk438569)* mutants.
DOI: https://doi.org/10.7554/eLife.29257.009

frequently detached from the PHC-AVG axon bundle (*Figure 2B and E*), whereas in *rig-6* mutants (either *ok1589* or *gk438569*), both HOA and PHC axons were detached from AVG, but the fasciculation of HOA and PHC was not affected (*Figure 2C and E*, and *Figure 2—figure supplement 2*). The double mutants of *casy-1* and *rig-6* showed an additive phenotype, in which all three axons were disassociated (*Figure 2D and E*). Taken together, the results indicate that *casy-1* and *rig-6* are required for proper wiring of the HOA-AVG-PHC circuit, but may function differently to achieve this goal.

## Axon fasciculation of HOA and AVG is promoted by PHC

As axon outgrowth and fasciculation are developmental processes, it is possible that the axon fasciculation defects we observed in mutants are affected by developmental timing of axon generation. Therefore we observed axon extension of the three neurons during male development. AVG grows its axon in the embryo and pioneers the right tract of the ventral nerve cord (*Durbin, 1987*). During the fourth larval stage (L4), in males but not hermaphrodites, PHC extends a process that follows along the axon of AVG (*Figure 2F and G*). Later, during the L4-adult transition, HOA extends its axon along the PHC-AVG bundle (*Figure 2F and G*). Thus, it is possible that axonal contact between HOA and AVG is promoted by PHC.

To test this possibility, we ablated PHC or HOA and examined axon fasciculation of the remaining neurons. When PHC was ablated, axonal contact between HOA and AVG was compromised, showing a fasciculation defect similar to that observed in *casy-1* and *rig-6* mutants (*Figure 2H and I*). However, PHC-AVG or HOA-PHC axonal contact was not affected by HOA or AVG ablation, respectively (*Figure 2H and I*). These results suggest there is an affinity between HOA and PHC and raise the possibility that in *rig-6* mutants, the primary defect is loss of adhesion between PHC and AVG, while HOA separates from AVG because of adhesion to PHC.

## CASY-1 and RIG-6 act in postsynaptic AVG for axon fasciculation

Next, we determined where CASY-1 and RIG-6 act to mediate axon fasciculation. Cell-specific rescue experiments showed that full-length *casy-1* cDNA when expressed in postsynaptic AVG, but not presynaptic HOA or PHC, rescued axon fasciculation defects in *casy-1* mutants (*Figure 3A*). Similarly, *rig-6* cDNA only when expressed in AVG rescued fasciculation defects of *rig-6* mutants (*Figure 3B*).

We also examined protein expression of CASY-1 and RIG-6 in AVG using transgenic animals with YFP-tagged cDNAs. In L4 males, right before synapse formation among HOA-AVG-PHC, CASY-1::YFP localized throughout the AVG axon including at potential synaptic sites in the pre-anal ganglion as well as in the anterior cell body (nucleus and cytoplasm) (*Figure 3C*). YFP::RIG-6 was also localized to the axon, but in the cell body was mainly cytoplasmic (*Figure 3D*). Taken together, both CASY-1 and RIG-6 function in postsynaptic AVG, possibly by transmembrane adhesion activity of axonally localized proteins, to make an axon fascicle with the presynaptic neurons.

## Extracellular domains required for CASY-1 and RIG-6 function

To gain insight into the functional domains and binding partners of CASY-1 and RIG-6, we generated transgenic animals expressing a series of deletion constructs for the extracellular domains of the two proteins in AVG and tested them for axon fasciculation. CASY-1 contains two cadherin domains and a Laminin, Neurexin, Sex hormone-binding globulin (LNS) domain in its extracellular region (*Figure 4A*). The full-length (Full) or two cadherin domains-deleted (ΔCads) transgene rescued the fasciculation defect of *casy-1* mutants. However, the LNS domain-deleted (ΔLNS) or all extracellular domains-deleted (ΔNt) transgene failed to rescue the defect (*Figure 4A and B*). Thus, the LNS domain of CASY-1 is essential for axon fasciculation, but the cadherin domains are dispensable for this function.

RIG-6 has six Ig domains and four fibronectin type III (FN[III]) domains in its extracellular region (*Figure 4C*). The full-length (Full) or four FN[III] domains-deleted (ΔFN[III]) transgene rescued the fasciculation defect of *rig-6* mutants, whereas Ig domains-deleted (Δ1-3Ig and Δ4-6Ig) transgenes could

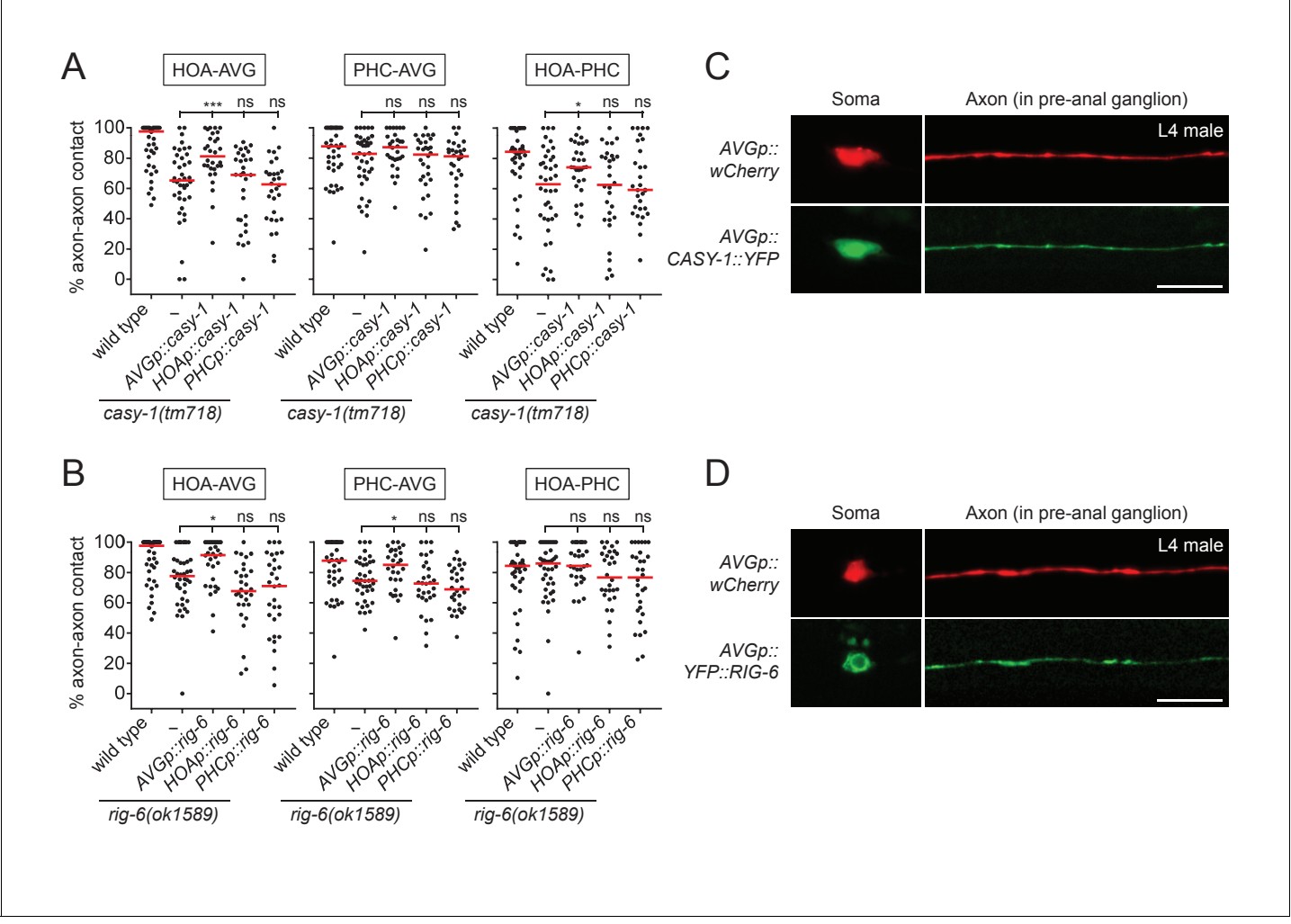

**Figure 3.** CASY-1 and RIG-6 act in postsynaptic AVG for axon fasciculation. (**A and B**) Percentage of axon-axon contact between each neuronal pair in animals expressing a full-length *casy-1* (**A**) or *rig-6* cDNA (**B**) in postsynaptic AVG or presynaptic HOA or PHC in a respective mutant background (n = 30). The data for wild type and *casy-1* and *rig-6* mutants are identical to those in *Figure 2* and are shown for comparison. Each dot represents individual animal. Red bar represents the median. *p<0.05; ***p<0.001; ns, not significant (by Mann-Whitney test). For the data and statistics, see *Figure 3—source data 1*. (**C and D**) Subcellular localization of YFP-tagged CASY-1 (**C**) or RIG-6 (**D**) in AVG (green) with the cytoplasmic marker wCherry (red) in L4 males. Scale bars, 20 μm. *AVGp*, *inx-18p* (*Oren-Suissa et al., 2016*); *HOAp*, *eat-4pA* (this study; see *Figure 1—figure supplement 1* for expression); *PHCp*, *eat-4p11* (*Serrano-Saiz et al., 2017a*).

DOI: https://doi.org/10.7554/eLife.29257.011

The following source data is available for figure 3:

**Source data 1.** Source data for *Figure 3*.
DOI: https://doi.org/10.7554/eLife.29257.012

not (*Figure 4C and D*), indicating that the Ig domains of RIG-6 are required for axon fasciculation. Taken together, these results suggest that the LNS domain of CASY-1 and the Ig domains of RIG-6 are functionally responsible for axon fasciculation, possibly by acting as the binding sites for trans-interacting partners.

## Neurexin-related BAM-2 is a presynaptic binding partner of CASY-1

Our finding of HOA-AVG axon defasciculation in *casy-1* mutants (*Figure 2*) suggests that a presynaptic binding partner may reside in HOA to interact with CASY-1 expressed by AVG. To identify such adhesion factors, we examined mutants for the cell adhesion genes expressed in HOA for defects in HOA presynaptic puncta. Among null mutants for the five genes expressed in HOA in our

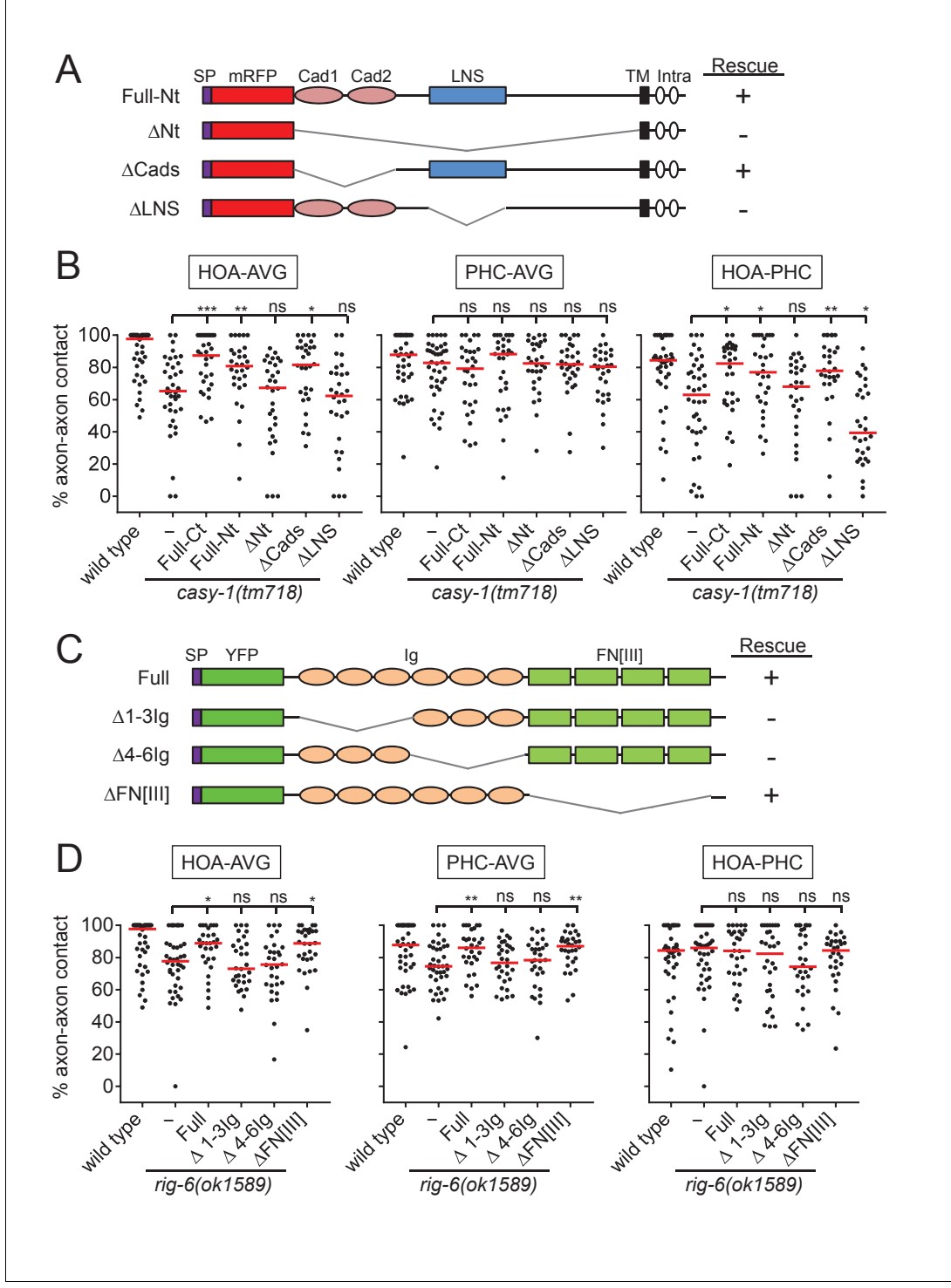

**Figure 4.** Structure-function analysis for CASY-1 and RIG-6. (**A and C**) Domain structure of mRFP::CASY-1 and its deletion constructs (**A**) or of YFP::RIG-6 and its deletion constructs (**C**) with the summary of rescue activity when expressed in AVG of *casy-1* or *rig-6* mutant animals. SP, signal peptide; Cad, cadherin domain; LNS, laminin neurexin sex hormone binding protein domain; TM, transmembrane; Intra, intracellular domain; Ig, immunoglobulin domain; FN[III], fibronectin type III domain. (**B and D**) Percentage of axon-axon contact between each neuronal pair in animals expressing the indicated deletion constructs for CASY-1 (**B**) or RIG-6 (**D**) in AVG of *casy-1* or *rig-6* mutant animals (n = 30). Note that Full-Ct (CASY-1::YFP) in (**B**) and Full (YFP::RIG-6) in (**D**) were the same transgenes used for protein localization in *Figure 3*, and rescued axon fasciculation defects in mutants. The data for wild type and *casy-1* and *rig-6* mutants are the same as shown in *Figure 2*. Each dot represents individual

*Figure 4 continued*

animal. Red bar represents the median. *p<0.05; **p<0.01; ***p<0.001; ns, not significant (by Mann-Whitney test). For the data and statistics, see *Figure 4—source data 1*.

DOI: https://doi.org/10.7554/eLife.29257.013

The following source data is available for figure 4:

**Source data 1.** Source data for *Figure 4*.

DOI: https://doi.org/10.7554/eLife.29257.014

expression dataset (*Figure 1E*), *bam-2(cy6)* mutants were identified as showing a gap phenotype similar to the *casy-1* mutant defect (*Figure 5A and B*, and *Figure 5—figure supplement 1*).

BAM-2 is a neurexin-related transmembrane protein that regulates axonal branch extension of hermaphrodite-specific VC ventral cord motorneurons in *C. elegans* (*Colavita and Tessier-Lavigne, 2003*). In a previous study, a CASY-1 homolog, calsyntenin-3, was shown to be an α-neurexin inter-acting partner, raising the possibility that BAM-2 is a presynaptic binding partner of CASY-1 (*Pettem et al., 2013*). Animals lacking *bam-2* showed axon fasciculation defects similar to the phenotype of *casy-1* mutants (*Figure 5C and D*). Double mutation of *casy-1* and *bam-2* did not enhance the axon fasciculation defect, suggesting that these genes act in the same genetic pathway (*Figure 5C and D*). Cell-specific rescue showed that full-length *bam-2* cDNA expressed in HOA, but not PHC, rescued axon fasciculation defects in *bam-2* mutants (*Figure 5E*). Finally, we tested protein interaction of CASY-1 and BAM-2 in a co-immunoprecipitation assay and found that Flag-tagged CASY-1 binds to V5-tagged BAM-2 (*Figure 5F*). In contrast, NRX-1, a canonical neurexin protein in *C. elegans*, did not bind to CASY-1 (*Figure 5—figure supplement 2*). These results suggest that BAM-2 is a functional presynaptic binding partner of CASY-1 for axon fasciculation.

## A short isoform of SAX-7/L1CAM is a presynaptic binding partner of RIG-6

The *rig-6* mutant axon fasciculation phenotype together with the PHC ablation result (*Figure 2*) suggest that a binding partner may function in PHC to interact with RIG-6 acting in AVG. Accordingly, we examined HOA presynaptic puncta in null mutants for the five cell adhesion genes expressed in PHC and found *sax-7(nj48)* showed the gap phenotype (*Figure 6A and B*, and *Figure 5—figure supplement 1*).

SAX-7/L1CAM, a transmembrane protein with six Ig and five FN[III] domains in its extracellular region, is required in *C. elegans* for maintenance of neuronal position including the cell body and axon (*Zallen et al., 1999*; *Sasakura et al., 2005*; *Pocock et al., 2008*). It has two isoforms – a long form with six Ig domains (SAX-7L) and a short form with four Ig domains (SAX-7S), where SAX-7S is shown to have more adhesive activity for cellular contact (*Sasakura et al., 2005*; *Pocock et al., 2008*). Some of the L1 family proteins including L1CAM are known to interact in trans with the RIG-6 ortholog contactin (reviewed in *Shimoda and Watanabe, 2009*). Several lines of evidence suggest that SAX-7S is a trans-binding partner of RIG-6 for axon fasciculation. First, in *sax-7(nj48)* mutants, the HOA and PHC axons were detached from AVG, but the HOA-PHC fasciculation was intact, which is similar to the phenotype of *rig-6* mutants (*Figure 6C and D*). Second, the double mutant phenotype of *rig-6;sax-7* animals was similar to that of *rig-6* or *sax-7* single mutants, suggesting that they act genetically in the same pathway (*Figure 6C and D*). Third, in cell-specific rescue experiments, the short form SAX-7S expressed in PHC, but not in HOA, rescued axon fasciculation defects in *sax-7* mutants, whereas SAX-7L failed to rescue both in PHC and HOA (*Figure 6E*). It is currently unknown if there are differences in expression of SAX-7S and SAX-7L, as different regulatory elements for the two isoforms have not yet been described. Finally, co-immunoprecipitation assay showed that Flag-tagged RIG-6 binds to V5-tagged SAX-7S (*Figure 6F*).

We observed above that PHC ablation disrupted the HOA-AVG axon fascicle (*Figure 2H and I*). We asked whether the loss of PHC could be compensated by restoring the interaction of SAX-7S with RIG-6 through expression of SAX-7S in HOA. SAX-7 protein was detected in PHC, but not in HOA or AVG, when we examined a fosmid-based reporter to better define the neurons expressing *sax-7* (*Figure 6—figure supplement 1*). Indeed, we found that the ectopic expression of SAX-7S in HOA rescued the fasciculation defects caused by PHC ablation (*Figure 6G and H*). Moreover, a

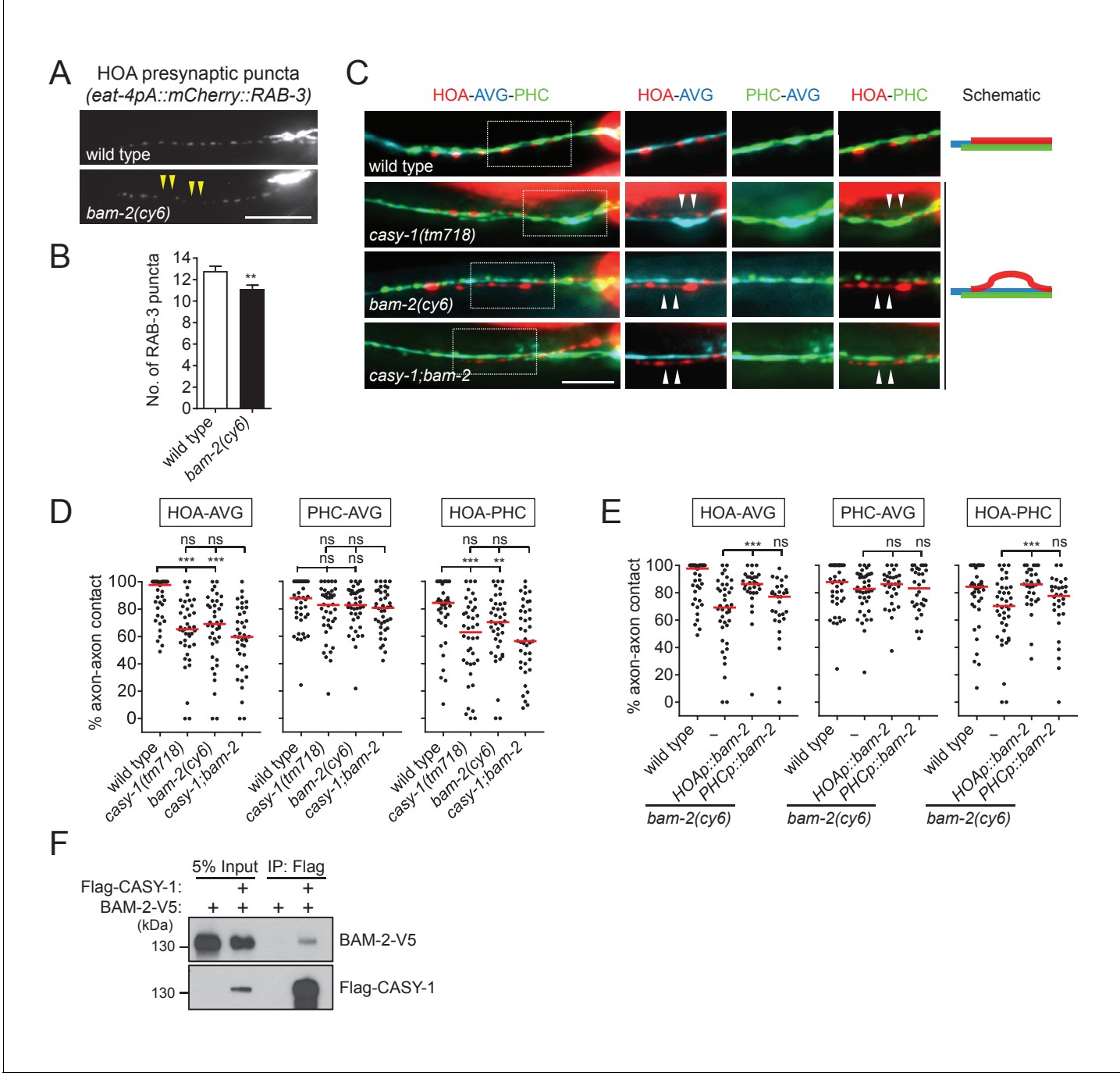

**Figure 5.** BAM-2 interacts with CASY-1. (**A**) Distribution of a mCherry-tagged presynaptic marker RAB-3 in HOA axon of wild type or *bam-2(cy6)* mutants. Arrowheads indicate gaps between the presynaptic puncta. (**B**) Number of mCherry::RAB-3 puncta in *bam-2* mutants was counted and compared to wild type (n = 30). Error bars are SEM. (**C**) Images of axon placement of HOA, AVG and PHC in *bam-2* and *casy-1;bam-2* double mutants. Axon fasciculation between each neuronal pair in the dashed box region and schematic of axon fasciculation are shown on the right. Arrowheads indicate the region where two axons are detached from each other. (**D**) Percentage of axon-axon contact between each neuronal pair in wild type or mutant animals (n = 40). Each dot represents individual animal. Red bar represents the median. (**E**) Percentage of axon-axon contact between each neuronal pair in animals expressing a full-length *bam-2* cDNA in HOA or PHC in *bam-2* mutant background (n = 30). In (**C–E**), the data for wild type and *casy-1* mutants are the same as shown in *Figure 2*. (**F**) Co-immunoprecipitation of CASY-1 and BAM-2. Scale bars, 20 μm. **p<0.01; ***p<0.001; ns, not significant (by Mann-Whitney test). For the data and statistics, see *Figure 5—source data 1*.

DOI: https://doi.org/10.7554/eLife.29257.015

The following source data and figure supplements are available for figure 5:

*Figure 5 continued*

**Source data 1.** Source data for *Figure 5*, *Figure 5—figure supplement 1*, and *Figure 5—figure supplement 2*.
DOI: https://doi.org/10.7554/eLife.29257.018
**Figure supplement 1.** HOA presynaptic puncta phenotypes of mutants for cell adhesion genes.
DOI: https://doi.org/10.7554/eLife.29257.016
**Figure supplement 2.** NRX-1 does not interact with CASY-1.
DOI: https://doi.org/10.7554/eLife.29257.017

similar rescuing effect was observed in a *casy-1* mutant, but not in a *rig-6* mutant (*Figure 6G and H*), further supporting the conclusion that SAX-7S binds to RIG-6 and this binding mediates axon fasciculation.

## Cell adhesion gene function is required for male behavior

What are the behavioral consequences of the connectivity defects in the circuit described above? The *C. elegans* male displays a stereotyped behavioral sequence during mating: it responds to contact with the hermaphrodite, backs with its tail pressed along the hermaphrodite body, turns at the end to the opposite side, locates its tail at the vulva, and inserts its copulatory spicules for insemination (*Liu and Sternberg, 1995*). The male-specific HOA sensory neuron is required for the vulva location step of this sequence (*Liu and Sternberg, 1995*). Recently, it has been found that genetic or laser ablation of AVG or PHC also leads to defects in vulva location (*Oren-Suissa et al., 2016*; *Serrano-Saiz et al., 2017a*). Thus, we anticipated that the cell adhesion genes required for the HOA-AVG-PHC circuit might affect the vulva location behavior of the male. We examined mating behavior of mutant males for *casy-1*, *rig-6*, *bam-2*, and *sax-7*, and all exhibited vulva location defects (*Figure 7A*). Double mutants (*casy-1;rig-6*, *casy-1;bam-2*, and *rig-6;sax-7*) also showed similar behavioral defects. Cell-autonomous expression of the cDNAs in the corresponding mutants partially restored vulva location behavior (*Figure 7A*). These defects appear to be specific to the vulva location step, since the male's ability to respond to hermaphrodite contact, the first step of the sequence, was not affected, with the exception of the *sax-7* mutation (*Figure 7B*). *sax-7* is thought to mediate multiple aspects of nervous system function including axon guidance, neuronal positioning, and dendrite development (*Zallen et al., 1999*; *Sasakura et al., 2005*; *Pocock et al., 2008*; *Dong et al., 2013*; *Salzberg et al., 2013*). In our expression analysis, *sax-7* was also expressed in some of the ray sensory neurons that are known to function in the response step (data not shown), raising the possibility that *sax-7* function in ray neurons is required for the response behavior. Interestingly, expression of SAX-7S in PHC partially rescued defects of *sax-7* mutants in the response to contact (*Figure 7B*), suggesting that response behavior also requires *sax-7* function in PHC. Taken together, these results suggest that the cell adhesion proteins described above regulate male mating behavior by mediating the formation of a normally responsive sensory circuit.

## Discussion

### The specification of HOA > AVG connectivity

We studied requirements for chemical synapse formation between the male-specific sensory neuron HOA and the sex-shared interneuron AVG. We found that three independent pathways appear to promote this connectivity. One involves interactions between cell surface proteins on the processes of HOA and AVG. A second involves another sensory neuron that is also postsynaptic to HOA, PHC, and interactions between cell surface proteins on PHC and AVG. A third pathway we did not identify by itself gives an approximately 50% level of fasciculation and synapse formation between HOA and AVG (*Figure 8*). Synapse formation between HOA and AVG takes place during the late period of the last, L4, larval stage, when HOA sends out a process along AVG. PHC grows a process fasciculated to AVG before HOA. The role of PHC in promoting HOA fasciculation to AVG is explained by an interaction between HOA and PHC, the molecular basis of which remains to be identified.

The involvement of a third cell such as PHC in promoting synapse formation appears to be a recurring mechanism in *C. elegans* synaptogensis. Epithelial cells of the vulva promote formation of neuromuscular junctions between the HSN motor neuron and vulval muscles (*Shen and Bargmann,*

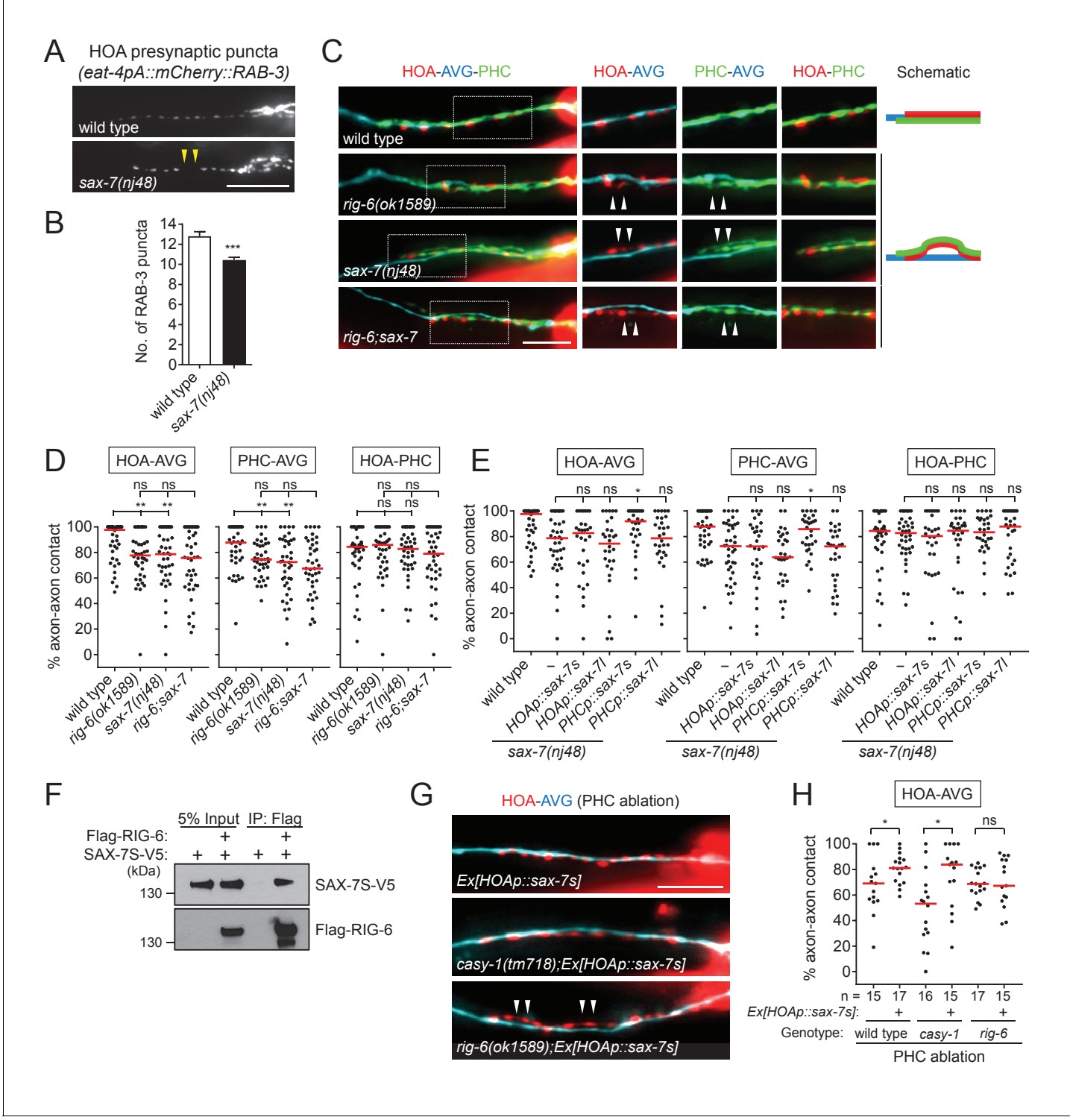

**Figure 6.** SAX-7 interacts with RIG-6. (**A**) Distribution of a mCherry-tagged presynaptic marker RAB-3 in HOA axon of wild type or *sax-7(nj48)* mutants. Arrowheads indicate gaps between the presynaptic puncta. (**B**) Number of mCherry::RAB-3 puncta in *sax-7* mutants was counted and compared to wild type (n = 30). Error bars are SEM. (**C**) Images of axon placement of HOA, AVG and PHC in *sax-7* and *rig-6;sax-7* double mutants. Axon fasciculation between each neuronal pair in the dashed box region and schematic of axon fasciculation are shown on the right. Arrowheads indicate the region where two axons are detached from each other. (**D**) Percentage of axon-axon contact between each neuronal pair in wild type or mutant animals (n = 40). Each dot represents individual animal. Red bar represents the median. (**E**) Percentage of axon-axon contact between each neuronal pair in animals expressing the short isoform *sax-7s* or long isoform *sax-7l* cDNA in HOA or PHC in *sax-7* mutant background (n = 30). In (**C–E**), the data for wild

*Figure 6 continued on next page*

*Figure 6 continued*

type and *rig-6* mutants are the same as shown in *Figure 2*. (F) Co-immunoprecipitation of RIG-6 and SAX-7S. (G) Images of axon placement of HOA and AVG in PHC-ablated animals expressing *sax-7s* cDNA in HOA (*Ex[HOAp::sax-7s]*). Arrowheads indicate the region where the HOA axon is detached from the AVG axon. (H) Percentage of axon-axon contact between HOA and AVG in PHC-ablated animals either expressing or not expressing *sax-7s* cDNA in HOA. The number of animals analyzed is indicated. The data for PHC-ablated wild type animals are the same as shown in *Figure 2I*. Scale bars, 20 μm. *p<0.05; **p<0.01; ***p<0.001; ns, not significant (by Mann-Whitney test). For the data and statistics, see *Figure 6— source data 1*.

DOI: https://doi.org/10.7554/eLife.29257.019

The following source data and figure supplement are available for figure 6:

**Source data 1.** Source data for *Figure 6*.
DOI: https://doi.org/10.7554/eLife.29257.021
**Figure supplement 1.** *sax-7* is expressed in PHC, but not in HOA or AVG.
DOI: https://doi.org/10.7554/eLife.29257.020

*2003*). The interaction involves binding between Ig-domain cell adhesion molecules SYG-2, expressed by the epithelium, and SYG-1, expressed by HSN (*Özkan et al., 2014*; *Shen et al., 2004*). Glial cells promote connection between AIY and RIA interneurons through the UNC-6/netrin pathway (*Colón-Ramos et al., 2007*). The combined involvement of more than two cells may be a significant way that a combinatorial complexity can be introduced into the mechanism of synaptic specificity.

The features of HOA > AVG connectivity we describe are in remarkable consonance with the features predicted in view of the complexity of the connectome and the structure of the nervous system. Each cell expresses multiple cell adhesion proteins. A subset of these are involved in formation of one particular synaptic connection. The cell adhesion proteins promote fasciculation, creating an extended neighborhood within which the neurons construct *en passant* synapses. Multiple pathways act independently and additively to create the wild type level of connectivity. We speculate that the pathway that can give rise to a 50% of wild type level of fasciculation even in the absence of the cell adhesion proteins we identified could be the cell lineage itself, which places cells at reproducible locations with reproducible sets of neighbors. As pointed out previously, unless a cell migrates away from its site of birth, it has a limited number of choices for process neighborhoods (*White, 1985*; *White et al., 1983*). Involvement of a third cell means lineage specification of that cell also contributes to specifying connectivity.

These features, which help to explain how the complex connectivity can be genetically encoded and robustly established, also explain why, in spite of decades of studies and analysis of thousands of mutations affecting the *C. elegans* nervous system and its behavior, the problem of overall specification of the connectome remains opaque. For connection of HOA to AVG, simultaneous removal of two sets of adhesion protein interactions reduces the level of fasciculation and hence synapse formation by only 50%. It is perhaps telling that some of the earliest uncoordinated mutations identified in *C. elegans*, which define the UNC-6/netrin guidance pathway and do in fact affect connectivity, affect cells that do not run in defined neighborhoods (*Brenner, 1973*; *Hedgecock et al., 1990*). Apart from these and some additional effects, in spite of the global role in guidance played by the UNC-6/netrin pathway, in UNC-6/netrin pathway mutants the nervous system remains intact and generally functional. Mutations in other widely-expressed genes for conserved pathways, such as the neurexin/neuroligin interaction and Slit/Robo signaling, have only mild effects on worm behavior. The nervous system is clearly a robust structure that is based on extensive genetic redundancy.

We surmounted this obstacle here by taking a candidate gene approach based on a comprehensive expression study. Two similar studies in *C. elegans* based on identifying the expression patterns of Ig-domain transmembrane protein genes followed by examination of mutants found similar axon guidance and fasciculation defects, including additive effects of multiple mutations, similar to the results described here (*Aurelio et al., 2002*; *Schwarz et al., 2009*). This approach may be more efficacious than forward genetics for deciphering the genetic code for nervous system connectivity.

## Synapse formation

For extensively-fasciculated neurons that communicate through *en passant* synapses, the problem of where to locate presynaptic structures must be solved. EM reconstruction shows that in one

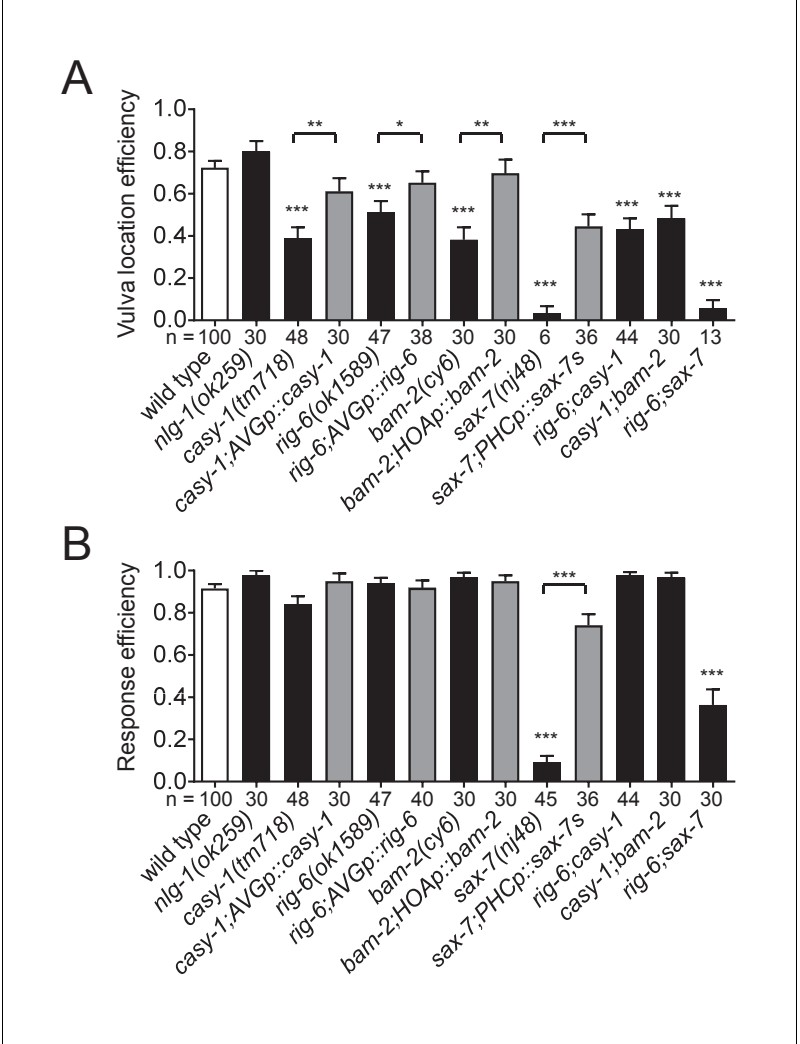

**Figure 7.** Cell adhesion genes required for male mating behavior. Vulva location efficiency (**A**) or response efficiency (**B**) in the indicated mutant males was measured and compared to wild type males. For rescue experiments, the same cell-autonomous rescuing transgenes for the circuit formation were used. The number of males analyzed is indicated below each column. Error bars are SEM. *p<0.05; **p<0.01; ***p<0.001 (by Mann-Whitney test). For the data and statistics, see *Figure 7—source data 1*.
DOI: https://doi.org/10.7554/eLife.29257.022

The following source data is available for figure 7:

**Source data 1.** Source data for *Figure 7*.
DOI: https://doi.org/10.7554/eLife.29257.023

individual, HOA was fasciculated with AVG over 30 microns. In the fasciculated region 46 presynaptic densities occur in 11 clusters relatively evenly spaced at intervals averaging 1.7 microns (*Supplementary file 1*). They occur at swellings in the HOA process that contained a mitochondrion. These swellings could be observed with a cytoplasmic marker (*Figure 1H*). What determines where these swellings, mitochondria, and presynaptic structures are located along the process?

We employed a fluorescently tagged protein of presynaptic vesicles, mCherry::RAB-3, to observe presynaptic structures. Absence of mCherry::RAB-3 puncta suggests that components of the presynaptic structure necessary to recruit synaptic vesicles are not assembled at discrete membrane locations. The formation of mCherry::RAB-3 puncta within HOA correlated with apparent fasciculation with AVG and required AVG contact, since in wild type they were absent or reduced in rare regions of process separation. For recruitment of presynaptic components, the contact signal did not require

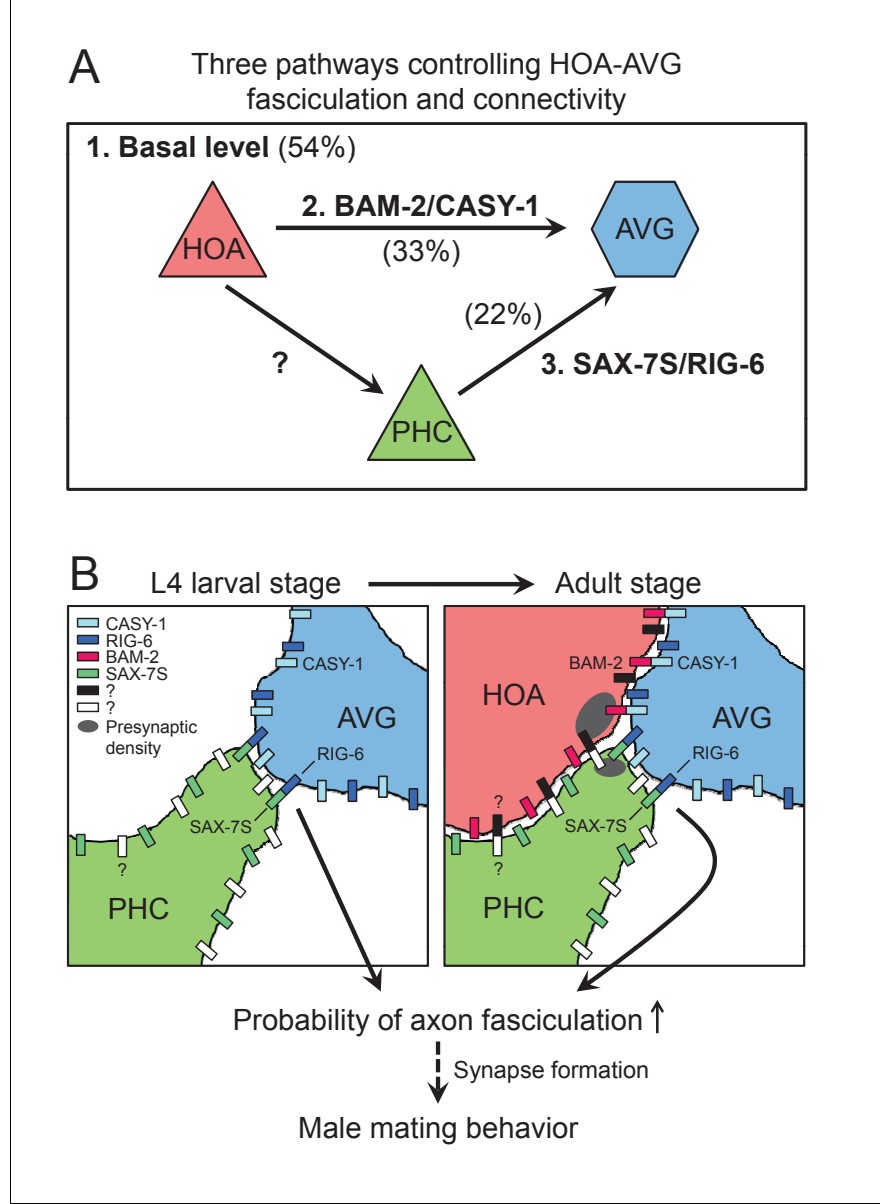

**Figure 8.** Model for function of cell adhesion protein interactions. (**A**) Three pathways additively act to achieve the wild type level of HOA-AVG axon fasciculation. The median percentages of HOA-AVG axon fasciculation in mutant and/or PHC ablation studies were used to estimate contribution of each pathway. (**B**) Schematic showing axons, cell adhesion protein interactions, and their putative function during the L4-adult transition in males. In L4 stage, the PHC axon contacts the AVG axon through RIG-6-SAX-7S interaction. In adult stage, close apposition of HOA and AVG membranes involves CASY-1-BAM-2 interaction, while unidentified protein interaction may mediate contact between HOA and PHC. These protein interactions regulate the probability of axon fasciculation and vulva location behavior possibly by controlling synapse formation between the neurons.
DOI: https://doi.org/10.7554/eLife.29257.024

the cell adhesion genes involved in promoting fasciculation, since in the mutants for these genes, puncta appeared normal in regions of contact (*Figure 1*). Of the 13 genes we examined, *oig-1* and *oig-8* may be necessary for the formation of puncta, because the mutants lacking them exhibited diffused puncta pattern (*Figure 5—figure supplement 1*). This diffused puncta phenotype is similar to that of animals lacking SYD-2/Liprin-α, which functions in formation of the presynaptic active zone (*Zhen and Jin, 1999*), suggesting that *oig-1* and *oig-8* may be involved in organizing presynaptic

structures. The identity of the contact signal and how it triggers assembly of presynaptic materials remains elusive.

## Functional conservation of cell adhesion protein interactions in synaptic partner recognition

We found that the cell adhesion genes required for promoting fasciculation between HOA and AVG are homologs of genes that have been shown to be involved in axon outgrowth and synapse formation in mammals. A similar finding has been made in wide-ranging prior studies of the *C. elegans* nervous system. Just as so much else of the molecular architecture of the nervous system is conserved across animals, so too, it appears, are the molecular tools that establish synaptic connectivity (*Bargmann, 1998*). Yet unlike many other aspects of neural function, such as the mechanisms underlying cellular electrical properties or synaptic transmission, connectivity differs profoundly from species to species and accounts for each species' repertoire of behaviors. The resolution of this apparent paradox, of course, is the employment of these generic cell-recognition and adhesion tools with different combinations in different places. The process is analogous to the specification of diverse cell types by a conserved set of transcription factors.

Our results show that CASY-1/calsyntenin binds to BAM-2/neurexin-related in trans to promote axon-axon adhesion. A similar protein interaction between calsyntenin-3 and α-neurexin has been shown to regulate synapse formation in mammals (*Pettem et al., 2013*). In addition, through domain analysis of CASY-1, we identified the LNS domain, but not the cadherin domains, is required for axon fasciculation, suggesting that the LNS domain may serve as a binding site for BAM-2 (*Figure 4*). A biochemical study has shown that the LNS domain of calsyntenin-3 alone is sufficient to bind α-neurexin (*Lu et al., 2014*). Interestingly, in contrast to CASY-1-BAM-2 interaction, we could not observe protein binding between CASY-1 and NRX-1 (a canonical neurexin in *C. elegans*) in the same experimental condition (*Figure 5—figure supplement 2*). This suggests that in *C. elegans* BAM-2 may exist as a related but diverged form of neurexin to act as a functional binding partner of CASY-1/calsyntenin. The conserved function of CASY-1/calsyntenin-BAM-2/neurexin-related interaction in neural adhesion and synapse formation seems to be maintained throughout evolution.

Like CASY-1-BAM-2 interaction, protein binding between RIG-6/contactin and SAX-7/L1CAM appears to be molecularly and functionally conserved. A number of protein interactions between contactins (six members: contactin-1–6) and L1 family proteins (four members: L1, CHL1, neurofascin and NrCAM) have been found in vertebrates (*Shimoda and Watanabe, 2009*; *Gennarini et al., 2017*). Trans-interactions between these proteins are thought to mediate axon outgrowth via contactin-2 binding to L1 or NrCAM (*Kuhn et al., 1991*; *Stoeckli et al., 1997*) and synapse formation via a putative interaction between contactin-5 and NrCAM (*Ashrafi et al., 2014*). Our domain analysis of RIG-6 indicates that the extracellular Ig domains, but not the FN[III] domains, are required for axon fasciculation, suggesting that RIG-6-SAX-7 interaction might be mediated via the Ig domains (*Figure 4*). Recent studies have revealed the importance of interactions between Ig domains in cell-cell recognition (*Özkan et al., 2013*, *2014*; *Carrillo et al., 2015*). Thus, it is possible that the Ig domains of RIG-6 interact directly with the Ig domains of SAX-7 to achieve adhesive function.

Direct in vivo evidence that these conserved adhesion factors have instructive roles in synaptic partner recognition is still lacking in any model organism. Resolution of this issue in the present context awaits solution of several technical issues, including selection of appropriate non-synaptic target neurons and the availability of specific promoters for selective expression in these cells.

## Implications for neuropsychiatric disorders

There is evidence for each of the four cell adhesion genes described in this study and their related or homologous genes that alterations affecting them are implicated in behavioral deficits or neuropsychiatric disorders. These include: (1) CASY-1/calsyntenin: expression level of calsyntenin-1 or −3 was altered in the human brain or mouse model of Alzheimer's disease (*Uchida et al., 2011*; *Vagnoni et al., 2012*). (2) RIG-6/contactin: loss of contactin-4 altered axon-target specificity in a visual circuit of mice and relevant behavior (*Osterhout et al., 2015*); some contactins (contactin-4–6) are genetically implicated in autism and other mental disorders (*Oguro-Ando et al., 2017*). (3) BAM-2/neurexin-related: neurexin is a well-known neural adhesion protein involved in autism and other cognitive diseases (*Südhof, 2008*). (4) SAX-7/L1CAM: L1 mutations are associated with diverse

neurological defects and mental retardation (*Fransen et al., 1997*). In our behavioral assays, loss of these adhesion factors disrupted vulva location behavior during male mating (*Figure 7*). Furthermore, we have shown that these proteins are required for formation of the corresponding sensory circuit. We propose that the cell adhesion proteins may contribute to behavioral performance, in part, through their adhesive functions in circuit assembly.

# Materials and methods

## *C. elegans* strains, molecular cloning and transgenes

All strains were maintained according to standard methods (*Brenner, 1974*). *him-5(e1490)* worms were used as reference strains to generate worm populations containing large numbers of males. Worms were grown at 20°C on standard nematode growth media (NGM) plates with OP50 *E. coli* (RRID:WB-STRAIN:OP50) as a food source. Mutant alleles used in this study are *casy-1(tm718) II*, *rig-6(ok1589) II*, *rig-6(gk438569) II*, *sax-3(ky123) X*, *bam-2(cy6) I*, *sax-7(nj48) IV*, *igcm-1(ok711) X*, *nlg-1 (ok259) X*, *oig-8(gk867223) II*, *oig-1(ok1687) III*, *lat-1(ok1465) II*, *ncam-1(hd49) X*, *wrk-1(ok695) X*, *zig-1(ok784) II*, *nrx-1(ok1649) V*, *nrx-1(wy778) V*, and *unc-31(e169) IV*.

Information on transgenic strains and DNA constructs used to generate transgenic animals is in *Supplementary file 2*.

## Analysis of electron micrograph

Electron micrograph images of N2Y series for adult male tail were analyzed using Elegance software (*Xu et al., 2013*). The analyzed sections in this study are from section #13836 (the anterior ends of the PHC axons) to section #14249 (the start point of the anteriorly-oriented HOA axon, which is right after the end point of the PCA axons). The images used in figures are from section #14160 (in *Figure 1B*) and #14193 (in *Figure 2—figure supplement 1*). Information on the section numbers, morphology of neurons, synaptic connectivity and images are accessible at WormWiring (http://wormwiring.org).

## Microscopy and image analysis

Worms were anesthetized with 10 mM sodium azide and mounted on 5% agar pads on glass slides. We used 1-day-old males unless otherwise indicated. Worms were observed with Nomarski or fluorescence microscopy (Zeiss Axio Imager.A1 or Z2), and images were acquired using the AxioCam camera (Zeiss) and processed using AxioVision (Zeiss). For *Figure 3C and D*, and *Figure 6—figure supplement 1*, Z1 Apotome was used to generate z-stacks, and one z-stack image was extracted from them. All figures were prepared using ImageJ software.

To measure the number of HOA presynaptic puncta, images were taken with the same amount of exposure time, and the mCherry::RAB-3 puncta that were anterior to the ending of PCA presynaptic signal were counted. The puncta size was measured using a line tool of ImageJ software. To obtain a percentage of axon-axon contact, we summed the lengths of contacting segments divided by total length of the shorter axon. The shorter axons were the HOA axon for HOA-AVG contact, the PHC axons for PHC-AVG contact, and the HOA axon (occasionally PHC) for HOA-PHC contact.

## Cell ablation

Cell ablation was performed using Pulsed UV, Air-Cooled, Nitrogen Laser System (Spectra-Physics, Santa Clara, CA) as described elsewhere (*Fang-Yen et al., 2012*). The L4 (for PHC or HOA ablation) or L1/L2 (for AVG ablation; the AVG axon was often visible after AVG ablation in L4 stage, which makes it difficult to interpret) stage males were anesthetized with 10 mM sodium azide and mounted on 5% agar pads on glass slides. Cells were identified with GFP (for PHCs), wCherry (for HOA), or TagBFP (for AVG) marker. After cell ablations, worms were recovered for 16–48 hr and the resulting adult worms were analyzed for imaging. Successful ablations were confirmed under the microscope, otherwise worms were discarded.

## Co-immunoprecipitation assay

HEK293T cells (RRID:CVCL_0063) were cultured in DMEM containing 10% fetal bovine serum, and maintained in 5% $CO_2$ at 37°C. Plasmids were transfected using Lipofectamine 2000 (Invitrogen)

according to manufacturer's instructions. The cells were lysed 24 hr after transfection with either RIPA lysis buffer (150 mM NaCl, 1% NP-40, 0.5% sodium deoxycholate, 0.1% SDS, 50 mM Tris-HCl [pH 7.4]) or single detergent lysis buffer (150 mM NaCl, 1% NP-40, 50 mM Tris-HCl [pH 8.0]), supplemented with HALT protease inhibitor cocktail (Thermo Scientific). The cell lysates were incubated with mouse monoclonal anti-FLAG M2 (Sigma-Aldrich, RRID:AB_259529) for 1 hr at 4°C and protein A/G agarose (Santa Cruz Biotechnology) was added to the lysates and then incubated overnight at 4°C. After washed twice with the lysis buffer, the agarose resin was suspended in 1x Laemmli sample buffer (Bio-Rad) and analyzed by immunoblot analysis using anti-FLAG M2 (1:1,000) or mouse monoclonal anti-V5 (1:3,000, Life Technologies, RRID:AB_2556564).

Interaction between Flag-RIG-6 and SAX-7S-V5 was observed when RIPA lysis buffer was used for cell lysis. Interaction between Flag-CASY-1 and BAM-2-V5 was seen only when single detergent lysis buffer was used. (In this mild lysis condition, although protein extraction was less efficient than in a harsh RIPA lysis condition, we could see that the protein interaction was preserved during the assay.) For interaction between Flag-CASY-1 and NRX-1-V5, no interaction was observed in both lysis conditions.

### Male mating behavior assay

Male mating behavior was monitored as described previously (*Liu and Sternberg, 1995*; *Peden and Barr, 2005*). L4 males were isolated from hermaphrotides and kept for 16–24 hr until they reached adult stage. Mating plates were freshly prepared with a 3 µl drop of 10x concentrated OP50 *E.coli* culture, and 10–15 1-day-old *unc-31(e169)* hermaphrodites were placed on the plates. Single adult males were added on the mating plates and analyzed for male behavior. Mating behavior was scored during 5 min or until the male fully stopped at the vulva, whichever occurred first. The numbers of male tail contacts (for response behavior) and passes or hesitations at the vulva (for vulva location behavior) were counted. Successful vulva location was defined as full stop of male tail at the vulva with a duration of 10 s. Response efficiency is calculated as 1/number of male tail contacts to respond, while vulva location efficiency is calculated as 1/number of encounters to stop (*Peden and Barr, 2005*).

### Statistical analysis

Statistical analysis was performed using GraphPad Prism software (version 7.01, RRID:SCR_002798). No statistical methods were used to compute sample size when this study was being designed. Statistical methods used are described in each figure legend.

## Acknowledgements

We would like to thank O Ivashikiv and B Suo for preparing transcriptional reporter lines; H Bülow, O Hobert, H Hutter, S Mitani, and K Shen for strains; C Díaz-Balzac, L García, Y Iino, Y Kohara, and M Lázaro-Peña for constructs; S J Cook for help in using Elegance software; and H Bülow and K Saied-Santiago for their help in conducting co-immunoprecipitation assays. Some strains were provided by the CGC, which is funded by NIH Office of Research Infrastructure Programs (P40 OD010440). Support was from NIH (R01 GM066897 and R01 MH112689) and the G Harold and Leila Y Mathers Charitable Foundation.

## Additional information

### Funding

| Funder | Grant reference number | Author |
| --- | --- | --- |
| National Institutes of Health | R01 GM066897 | Scott W Emmons |
| G Harold and Leila Y. Mathers Foundation | | Scott W Emmons |
| National Institutes of Health | R01 MH112689 | Scott W Emmons |

The funders had no role in study design, data collection and interpretation, or the decision to submit the work for publication.

## Author contributions
Byunghyuk Kim, Conceptualization, Formal analysis, Investigation, Writing—original draft, Writing—review and editing; Scott W Emmons, Conceptualization, Supervision, Funding acquisition, Writing—review and editing

## Author ORCIDs
Byunghyuk Kim [iD] http://orcid.org/0000-0002-7967-7406

## Decision letter and Author response
Decision letter https://doi.org/10.7554/eLife.29257.028
Author response https://doi.org/10.7554/eLife.29257.029

# Additional files

### Supplementary files
• Supplementary file 1. Neighborhood and synapse in the HOA-AVG-PHC circuit
DOI: https://doi.org/10.7554/eLife.29257.025

• Supplementary file 2. *C. elegans* strains, molecular cloning and transgenes
DOI: https://doi.org/10.7554/eLife.29257.026

• Transparent reporting form
DOI: https://doi.org/10.7554/eLife.29257.027

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
