## [Decision Letter]

Thank you for submitting your article "Multiple conserved cell adhesion protein interactions mediate neural wiring of a sensory circuit in *C. elegans*" for consideration by *eLife*. Your article has been favorably evaluated by Eve Marder (Senior Editor) and three reviewers, one of whom is a member of our Board of Reviewing Editors. The reviewers have discussed the reviews with one another and the Reviewing Editor has drafted this decision to help you prepare a revised submission.

This manuscript addresses the question of how neurons are able to make precise synaptic connections within complex circuits utilizing a limited repertoire of cell recognition molecules. The authors utilize a relatively simple three-neuron sensory circuit (AVG, PHC, and HOA) in male *C. elegans* to uncover that neural adhesion molecules with differential expression in neurons regulate fasciculation of their axons, which in turn impacts synapse formation between the neurons. Specifically, the authors define that the HOA neuron expresses BAM-2, which regulates axon fasciculation with AVG though interaction with CASY-1, and similarly, SAX-7S in PHC regulates axon fasciculation with AVG through interaction with RIG-6. The authors show an interesting neuronal hierarchy in regulation of axon fasciculation and perform cell-specific rescue experiments for the cell adhesion molecules in each neuron. The authors also identify the regions of the adhesion molecules required for the interactions they uncover, show direct binding using in vitro experiments, and lastly demonstrate that the cell adhesion molecules have an impact on a behavior relevant to the circuit. This work begins to elucidate the mechanisms by which neural adhesion molecules that seem to generally regulate synapse formation in neurons, are also able to promote precise synaptic connections between specific neurons.

However, there are a number of issues that the authors will need to address to strengthen their conclusion of the existence of a hierarchically acting, cell-type specific adhesion code.

Experimental:

1) The authors imply that the selective expression of these adhesion molecules in the three neurons instructs synaptic partnership. To demonstrate this point, they need to express these molecules in a neighboring non-synaptic partner neuron to show sufficiency of generating ectopic fasciculation and synapse. Loss of function effects could be due to many potential defects during development. Gain-of function manipulations make the model a lot stronger.

2) To solidify the argument of a hierarchical assembly of the PHC/HOA/AVG circuit, the authors need to also ablate AVG. Currently, they only ablate PHC and HOA.

3) The behavioral defects shown in Figure 7 are difficult to interpret without cell autonomous rescue experiments. Since the authors have already built these rescuing strains, they should at least test some of the strains to see if the AVG-PHC-HOA phenotype is responsible for the behavioral phenotypes.

4) The presentation of the expression pattern of the 4 relevant adhesion molecules is insufficient. The authors refer to their own unpublished expression analysis which, as some hints in the text suggest, appear to be based on transcriptional reporters. Two of those are shown in the context of demonstrating that these transcriptional reporters are not dimorphically expressed; this finding is uninterpretable if it based on transcriptional reporters only. It could easily be envisioned that sex-specifically occupied repressor sites are located outside the 5' region. Better reporter reagents are also critical for a proper interpretation of the elegant HOA-ectopic SAX-7S expression experiments in a PHC(-) or casy-6(-) background. Is this expression really ectopic (i.e. SAX-7S is not normally expressed in HOA) or do the authors merely overexpress a highly adhesive homophilic adhesion molecule that can compensate for the loss of other adhesive mechanisms? Taken together, the authors must use reporter strains in which the entire locus is tagged with gfp (either in the context of fosmid-based reporters or by CRISPR/Cas9 tagging).

5) It is not clear how sensitive the visual assay is in determining a "fasciculation" phenotype. The distance between neighboring axons is well below the resolution of light microscopy. When two axons appear to be "fasciculated" with each other on a two color image, they are not necessarily fasciculated. This is not a problem for the wild type control because that has been validated by EM. We understand that EM reconstruction is a major undertaking and it would not be reasonable to ask the authors to do EM on all the mutants. But it would be reasonable to have a better understanding about how sensitive their assays are. If the authors have strains available that label two axons in the nerve bundles that are separated by 1, 2 or 3 neurites the authors could corroborate that they can use their fluorescence images to determine that these axons are not fasciculating with each other. This is an important issue because it affects the interpretation of the whole paper.

Requested changes relating to data presentation and text:

1) Figure 1 showed about 12 puncta of HOA presynaptic terminals. The EM analyses showed 44 synapses to AVG and 46 to PHC. Why are these numbers so far apart from each other?

2) "the HOA axon forms en passant dyadic chemical synapses onto AVG and PHC". However, from Figure 1, it looks like PHC(L) labeled as "4" in the EM contains both synaptic vesicles and active zone structures. Why is this not a presynaptic terminal?

3) The in vitro immunoprecipitation experiment testing BAM-2 CASY-1 physical interaction is not as convincing as the RIG-6 SAX-7 pair (which looks good); I mention this since the ratio of signal between BAM-2-V5 and FLAG-CASY-1 IP (Figure 5), by eyeball, is very skewed and different from the SAX-7S-V5 AND Flag-RIG-6 IP ratio (Figure 6); the immunoblot looks more exposed to visualize the BAM-2 band. One possibility is that the in vitro binding conditions that the authors used for the experiment might not be optimal for the BAM-2:CASY-1 pair, or that some of the proteins might be damaged during the purification process. The authors might discuss these possibilities in their Materials and methods section and probably also explain technical reasons why doing IP-pull downs on the reciprocal partner (i.e. IP CASY-1 and detecting for BAM-2; IP RIG-6 and detecting for SAX-7) were not also done in the experiments.

4) There is no proper quantification of the observations shown in Figure 2. Panels in Figure 2 should include a landmark for comparison of each axon projection (rectum etc.).

5) The quality of the critical image in Figure 2 – the defasciculation of AVG and HOA upon PHC ablation is poor.

6) There is not panel H which supposedly shows the quantification of the ablation data.

7) It should be addressed in the text why AVG ablation was not performed in parallel with PHC and HOA ablations.

8) The authors should make it clear in the text that experiments in Figure 4 are expressed in a cell-specific manner.

9) It should be addressed if there are differences in expression of sax-7 long and short isoforms?

10) Results – less should be changed to fewer "exhibited gaps containing less or smaller puncta".

11) The following statement is confusing: "Throughout the paper the hypomorph was used because of its ease of genetics. Its phenotype is the same as the null. " How do the authors know the phenotype is the same if they did not use the null?

12) The following statement is incorrect: "In a previous study, an interacting partner of a CASY-1 homolog, calsyntenin-3, was shown to be α-neurexin, raising[…].". Calsyntenin is an α-neurexin interacting protein, it is not α neurexin.

---

## [Author Response]

However, there are a number of issues that the authors will need to address to strengthen their conclusion of the existence of a hierarchically acting, cell-type specific adhesion code.Experimental:1) The authors imply that the selective expression of these adhesion molecules in the three neurons instructs synaptic partnership. To demonstrate this point, they need to express these molecules in a neighboring non-synaptic partner neuron to show sufficiency of generating ectopic fasciculation and synapse. Loss of function effects could be due to many potential defects during development. Gain-of function manipulations make the model a lot stronger.

We do not mean to imply that expression of these adhesion molecules *by themselves* is sufficient to cause adhesion. Most likely it is not, but rather these molecules act in concert with other factors, possibly in complexes. Throughout the manuscript we use the word “promote” to describe their role in adhesion, which is simply to say they are necessary without saying how.

We do not feel that a gain-of-function experiment, showing an effect of expression in other neurons, changes this perspective. An effect in another pair of neurons might occur because the same set of necessary co-factors are present in those cells. Lack of effect could be because the other factors are not there. In other words, cellular context is undoubtedly important and interpretation of a “sufficiency” experiment is problematic. If expression in another context is not sufficient, that does not take away from the evidence that in the present context these proteins are necessary. Though undoubtedly there may be other explanations, the simplest explanation for their role in “promoting” adhesion is that they bind to each other.

Having said all this, we agree with the reviewer’s point that gain-of-function experiments strengthen our argument on the roles of cell adhesion proteins. To address whether the adhesion factors can instruct synaptic partnership, we initially tried to find a good candidate for a neighboring non-synaptic partner neuron, but we confronted two obstacles.

The first obstacle was to select an appropriate target neuron for “a non-synaptic partner” that is testable. As discussed in the response to the Experimental issue #5, closely running axons within 1~3 neurite distance frequently touch each other, making it difficult to interpret the results, that is, it could be difficult to see if these contacts were *increased*. As for axons normally separated by long distance, it is hard to assess whether some kinds of inductive effects work or not, if there are any, because we do not know the effective ranges of the adhesion proteins.

The second problem was the availability of specific promoters that can drive expression in the neurons we want to examine. We knew that many selective drivers known in hermaphrodites have additional expression in the male tail (e.g. a subset of male-specific neurons or muscles) through our extensive expression analyses on cell adhesion genes and many other genes. Even so although we may identify a certain target neuron that is testable (i.e. non-synaptic partner with a close neighborhood), it is challenging to find selective drivers showing unique expression in that neuron in the male tail.

We did find one promising candidate and performed a gain-of-function test, but it was unsuccessful. The technical difficulties described above are well exemplified in our preliminary test of ectopic expression in the HOB neuron in the hope of finding some recognizable inductive effects. HOB has synaptic partnership with HOA and PHC only in a restricted “<”-shaped axonal region. The *pkd-2* promoter has been well known to drive expression selectively in HOB (additionally in the B-type ray neurons). In this experiment, we ectopically expressed CASY-1 and RIG-6 in HOB in a *casy-1;rig-6* double mutant background and looked for re-routing of the anteriorly-oriented axons of HOA and PHC toward the HOB axon instead of following the AVG axon. However, we could not see such effects (see Author response image 1).

This result is not decisive. First, our initial prediction of the re-routing may be too rough. Even though the HOA and PHC axons have contacts with the HOB axons in one limited region, their anterior axons are still far from the HOB axon. Due to this long distance, the adhesion factors expressed in HOB may not reach factors on the presynaptic neurons, HOA and PHC. Second, the promoter utilized (*pkd-2p*) is very selective in expression, but is not specific to HOB. There remains the possibility that the adhesion factors ectopically expressed in a subset of the B-type ray neurons’ axons (e.g. R1B whose axon runs near and has significant contacts with the AVG axon) may have some effects. Taken together, we think that currently we do not have appropriate tools to assess this problem. We have described these difficulties briefly in the Discussion.

2) To solidify the argument of a hierarchical assembly of the PHC/HOA/AVG circuit, the authors need to also ablate AVG. Currently, they only ablate PHC and HOA.

We have conducted AVG ablation. There was no effect in HOA-PHC fasciculation after the ablation. We have added this result in Figure 2 and in the relevant text.

3) The behavioral defects shown in Figure 7 are difficult to interpret without cell autonomous rescue experiments. Since the authors have already built these rescuing strains, they should at least test some of the strains to see if the AVG-PHC-HOA phenotype is responsible for the behavioral phenotypes.

As suggested, we have conducted cell-autonomous rescue experiments for the four mutants. The results show that vulva location defects in the mutants are partially rescued by cell-autonomous expression of the corresponding genes. Accordingly, we have incorporated these data into Figure 7.

4) The presentation of the expression pattern of the 4 relevant adhesion molecules is insufficient. The authors refer to their own unpublished expression analysis which, as some hints in the text suggest, appear to be based on transcriptional reporters. Two of those are shown in the context of demonstrating that these transcriptional reporters are not dimorphically expressed; this finding is uninterpretable if it based on transcriptional reporters only. It could easily be envisioned that sex-specifically occupied repressor sites are located outside the 5' region. Better reporter reagents are also critical for a proper interpretation of the elegant HOA-ectopic SAX-7S expression experiments in a PHC(-) or casy-6(-) background. Is this expression really ectopic (i.e. SAX-7S is not normally expressed in HOA) or do the authors merely overexpress a highly adhesive homophilic adhesion molecule that can compensate for the loss of other adhesive mechanisms? Taken together, the authors must use reporter strains in which the entire locus is tagged with gfp (either in the context of fosmid-based reporters or by CRISPR/Cas9 tagging).

We thank the reviewer for raising this issue. For *casy-1* and *rig-6* expression, we agree with the reviewer’s point that it is not interpretable in terms of sexually dimorphic expression due to possible exclusion of repression sites. We have removed Figure 3—figure supplement 1 and are pursuing to generate CRISPR lines for endogenous expression of *casy-1* and *rig-6* in order to understand their sex-specific regulation.

For *sax-7* expression, we have used a *sax-7* fosmid reporter (ddIs290) to examine its expression in PHC, HOA, and AVG. The results indicate that *sax-7* is selectively expressed in PHC and are now presented as Figure 6—figure supplement 1.

5) It is not clear how sensitive the visual assay is in determining a "fasciculation" phenotype. The distance between neighboring axons is well below the resolution of light microscopy. When two axons appear to be "fasciculated" with each other on a two color image, they are not necessarily fasciculated. This is not a problem for the wild type control because that has been validated by EM. We understand that EM reconstruction is a major undertaking and it would not be reasonable to ask the authors to do EM on all the mutants. But it would be reasonable to have a better understanding about how sensitive their assays are. If the authors have strains available that label two axons in the nerve bundles that are separated by 1, 2 or 3 neurites the authors could corroborate that they can use their fluorescence images to determine that these axons are not fasciculating with each other. This is an important issue because it affects the interpretation of the whole paper.

We thank the reviewer for raising this issue. It would be ideal if we could find two axons that run in parallel but are separated by one neurite in EM, and then examine those using fluorescence images. Unfortunately, it is hard to find such an axon pair in EM, because most axons that run close together (1~3 neurites distance) frequently “touch” and “detach” throughout their lengths. Instead, we have focused on the fasciculation of PHCL and PHCR axon pair, which defasciculate *with one-neurite distance separation* in EM over 30% of their length. We have compared this with fluorescence images and found similar levels of defasciculation (~29%). This result suggests that the fluorescent images are quite sensitive and has been presented in Figure 2—figure supplement 1 and in the relevant text.

Requested changes relating to data presentation and text:1) Figure 1 showed about 12 puncta of HOA presynaptic terminals. The EM analyses showed 44 synapses to AVG and 46 to PHC. Why are these numbers so far apart from each other?

The puncta are likely to represent ~11 clusters of presynaptic densities, rather than individual synapses (n=46). This was previously described in Discussion, and now we have repeated it in Results.

2) "the HOA axon forms en passant dyadic chemical synapses onto AVG and PHC". However, from Figure 1, it looks like PHC(L) labeled as "4" in the EM contains both synaptic vesicles and active zone structures. Why is this not a presynaptic terminal?

It is a presynaptic terminal of PHC. In the previous version of this manuscript, we intentionally focused on describing HOA, but now have added this fact in the text.

3) The in vitro immunoprecipitation experiment testing BAM-2 CASY-1 physical interaction is not as convincing as the RIG-6 SAX-7 pair (which looks good); I mention this since the ratio of signal between BAM-2-V5 and FLAG-CASY-1 IP (Figure 5), by eyeball, is very skewed and different from the SAX-7S-V5 AND Flag-RIG-6 IP ratio (Figure 6); the immunoblot looks more exposed to visualize the BAM-2 band. One possibility is that the in vitro binding conditions that the authors used for the experiment might not be optimal for the BAM-2:CASY-1 pair, or that some of the proteins might be damaged during the purification process. The authors might discuss these possibilities in their Materials and methods section and probably also explain technical reasons why doing IP-pull downs on the reciprocal partner (i.e. IP CASY-1 and detecting for BAM-2; IP RIG-6 and detecting for SAX-7) were not also done in the experiments.

We used RIPA buffer (harsh lysis condition) in an initial test and observed RIG-6/SAX-7S interaction, but neither CASY-1/BAM-2 nor CASY-1/NRX-1 interaction. We reasoned that harsh condition might inhibit some protein interactions, so we tested them using single detergent lysis buffer. This condition was less efficient in protein extraction from cells (reflected in the faint protein bands for CASY-1 and BAM-2) but seemed enough to preserve the protein interaction between CASY-1 and BAM-2. However, in both lysis conditions, CASY-1/NRX-1 interaction was negative. We have discussed this in Materials and methods section.

4) There is no proper quantification of the observations shown in Figure 2. Panels in Figure 2 should include a landmark for comparison of each axon projection (rectum etc.).

We have added quantification data in Figure 2. As a landmark, we have indicated the posterior end of the intestine in Figure 2.

5) The quality of the critical image in Figure 2 – the defasciculation of AVG and HOA upon PHC ablation is poor.

For better presentation, we have included a magnified image of the figure.

6) There is not panel H which supposedly shows the quantification of the ablation data.

In the previous version of this manuscript, the panel H was located next to the panel E, which may make it difficult to find. We have rearranged the Figure 2, and the quantification is now shown in Figure 2.

7) It should be addressed in the text why AVG ablation was not performed in parallel with PHC and HOA ablations.

We have conducted AVG ablation as described above in the experimental point #2.

8) The authors should make it clear in the text that experiments in Figure 4 are expressed in a cell-specific manner.

We have revised the text accordingly.

9) It should be addressed if there are differences in expression of sax-7 long and short isoforms?

This is an interesting question with no answer. Currently, no information is available for different regulatory elements for each isoform, which makes it difficult to address this question. We have explained this in the text.

10) Results – less should be changed to fewer "exhibited gaps containing less or smaller puncta".

We have revised the text accordingly.

11) The following statement is confusing: "Throughout the paper the hypomorph was used because of its ease of genetics. Its phenotype is the same as the null. " How do the authors know the phenotype is the same if they did not use the null?

In the previous version of this manuscript, we compared the phenotypes of the hypomorph with those of the null (in Figure 1—figure supplement 3 and Figure 2—figure supplement 2) and found that they are the same. However, to avoid misunderstanding, we have rewritten the text.

12) The following statement is incorrect: "In a previous study, an interacting partner of a CASY-1 homolog, calsyntenin-3, was shown to be α-neurexin, raising[…].". Calsyntenin is an α-neurexin interacting protein, it is not α neurexin.

We have revised the text to clarify the meaning of the sentence.